# Galectin 3 Deficiency Alters Chondrocyte Primary Cilium Formation and Exacerbates Cartilage Destruction via Mitochondrial Apoptosis

**DOI:** 10.3390/ijms21041486

**Published:** 2020-02-22

**Authors:** Narjès Hafsia, Marine Forien, Félix Renaudin, Delphine Delacour, Pascal Reboul, Peter Van Lent, Martine Cohen-Solal, Frédéric Lioté, Françoise Poirier, Hang Korng Ea

**Affiliations:** 1Université de Paris, BIOSCAR UMR 1132, Inserm, F-75010 Paris, France; narjes.hafsia@gmail.com (N.H.); marine.forien@aphp.fr (M.F.); felix.renaudin@inserm.fr (F.R.); martine.cohen-solal@inserm.fr (M.C.-S.); frederic.liote@aphp.fr (F.L.); 2UMR 7592 CNRS, Institut Jacques Monod, Univ. Paris Diderot, Sorbonne Paris Cité, F-75205 Paris, France; delphine.delacour@ijm.fr (D.D.); poirier@ijm.univ-paris-diderot.fr (F.P.); 3UMR 7365, CNRS-Université de Lorraine, IMoPA, F-54000 Vandœuvre-lés-Nancy, France; pascal.reboul@univ-lorraine.fr; 4Rheumatology Research and Advanced Therapeutics, Department of Rheumatology, Radboud University Medical Centre, 6500 HB Nijmegen, The Netherlands; peter.vanlent@radboudumc.nl; 5Service de Rhumatologie, Centre Viggo Petersen, AP-HP, hôpital Lariboisière, F-75010 Paris, France

**Keywords:** Galectin 3, primary cilium, chondrocyte, osteoarthritis, mechanical stress, apoptosis

## Abstract

Mechanical overload and aging are the main risk factors of osteoarthritis (OA). Galectin 3 (GAL3) is important in the formation of primary cilia, organelles that are able to sense mechanical stress. The objectives were to evaluate the role of GAL3 in chondrocyte primary cilium formation and in OA in mice. Chondrocyte primary cilium was detected in vitro by confocal microscopy. OA was induced by aging and partial meniscectomy of wild-type (WT) and *Gal3*-null 129SvEV mice (*Gal3^−/−^*). Primary chondrocytes were isolated from joints of new-born mice. Chondrocyte apoptosis was assessed by Terminal deoxynucleotidyl transferase dUTP nick end labeling (TUNEL), caspase 3 activity and cytochrome c release. Gene expression was assessed by qRT-PCR. GAL3 was localized at the basal body of the chondrocyte primary cilium. Primary cilia of *Gal3^−/−^* chondrocytes were frequently abnormal and misshapen. Deletion of *Gal3* triggered premature OA during aging and exacerbated joint instability-induced OA. In both aging and surgery-induced OA cartilage, levels of chondrocyte catabolism and hypertrophy markers and apoptosis were more severe in *Gal3^−/−^* than WT samples. In vitro, *Gal3* knockout favored chondrocyte apoptosis via the mitochondrial pathway. GAL3 is a key regulator of cartilage homeostasis and chondrocyte primary cilium formation in mice. *Gal3* deletion promotes OA development.

## 1. Introduction

Osteoarthritis (OA) is a multifaceted joint disease characterized by cartilage degradation, bone modifications and mild synovial inflammation. OA cartilage displays extracellular matrix and cell modifications including increased production of metalloproteases (MMPs) and aggrecanases (ADAMTS-4 and 5, A Disintegrin And Metalloprotease with ThromboSpondin-like repeat) and increased chondrocyte catabolic and hypertrophic differentiation [1,2]. Factors associated with OA development include genetics, sex, metabolic syndrome, obesity and diabetes, but aging and mechanical overload are the two most prominent risks [2,3,4]. However, the precise molecular mechanisms responsible for initiation or progression of OA remain to be elucidated.

Several reports have recently unveiled the importance of the primary cilium in cartilage physiology and pathology [5,6,7,8,9,10]. Primary cilium located at the chondrocyte surface has the capacity to sense multiple stimuli including mechanical stress and inflammatory cytokine stimulation [5,6,7,8,9,10,11,12]. Thus, stimulation with inflammatory interleukin 1β (IL-1β) increases the chondrocyte primary cilium length within minutes [10]. Furthermore, mechanical stimulation promotes primary cilium-driven intracellular Ca^2+^ mobilization and Indian Hedgehog (Ihh) signaling [5,11]. For instance, chondrocytes isolated from mice with mutant Polaris/intraflagellar transport (ITF) 88, a protein necessary for cilium formation, fail to increase intracellular Ca^2+^ under mechanical stimulation [11]. Moreover, loss of primary cilia in chondrocytes reduces cartilage mechanical properties in *Col2Cre/Ift88^ft/ft^* transgenic mice and promotes OA development characterized by increased expression of MMP13, ADAMTS5, COLX and RUNX2 [12,13]. Similarly, mice mutant for Bardet-Biedl syndrome 1 (*Bbs1^−/−^)*, *Bbs2^−/−^* or *Bbs6^−/−^*, a family of proteins involved in primary cilium formation/functions, develop OA-like cartilage abnormalities including proteoglycan loss, small surface fibrillations, reduced cartilage thickness and increased MMP13 expression [14,15].

The expression of galectin 3 (GAL3), a 30-kDa member of the galectin family of lectins, is increased in human OA versus normal cartilages [16,17]. GAL3 is multifunctional protein implicated in cellular interactions, cell differentiation, survival and death [18]. The role of GAL3 in OA has not been completely elucidated: intracellular GAL3 has an anti-apoptotic effect whereas extracellular GAL3 stimulates ADAMTS-5 production [19,20]. Importantly, recent work revealed the involvement of GAL3 in cilium biogenesis in two tissue types [21,22]. In epithelial renal cells, GAL3 is normally localized at the base of primary cilia, and its absence causing primary cilium abnormalities is associated with major defects in epithelial cell polarity [22]. In tracheal cells, GAL3 is a component of the basal-foot cap of motile cilia, and its absence causes perturbed ciliary organization, which provokes defects in mucus clearance [21]. Whether GAL3 is involved in chondrocyte primary cilium formation is unknown.

In this study, we used the *Gal3^−/−^* mouse to (1) directly assess the role of GAL3 in OA and (2) further explore the importance of GAL3 in chondrocyte primary cilium formation.

We observed that GAL3 was a key regulator of cartilage homeostasis in mice, particularly in response to mechanical stress induced by joint instability. GAL3 deletion promoted OA and altered chondrocyte functions including formation of primary cilium, catabolic activities and cell death.

## 2. Results

### 2.1. GAL3 Deficiency Leads to Spontaneous OA in 14-Month-Old Mice and Exacerbates Surgery-Induced OA in 3-Month-Old Mice

We first assessed the consequences of *Gal3* deletion in two OA models. The cartilages of old (14 months) and young (3 months) wild-type (WT) and *Gal3^−/−^* mice were compared after safranin-O staining of knee sections (Figure 1A and Figure 2A). The extent and intensity of safranin-O staining was similar between 3- and 14-month-old WT mice, so the thickness and the proteoglycan content of knee cartilage remained unchanged during aging. *Gal3^−/−^* mice, which did not have an apparent skeletal anomaly during adulthood, had same mean body weight as WT mice. Safranin-O staining of 3-month-old cartilage was indistinguishable from that of 3-month-old WT mice, but severe defects appeared with age. The knee cartilage of 14-month-old *Gal3^−/−^* mice showed a decrease in cartilage thickness with extensive loss of proteoglycan, associated with cartilage fissuration (Figure 1A). With the OARSI scoring system used to quantify these observations, we found a high mean score for 14-month-old *Gal3^−/−^* mice (4.3 (range 0–8.0), *n* = 5 mice) and a normal mean score for 14-month-old WT mice (0.7 (0–2.0), *n* = 7 mice, *p* = 0.014). We conclude that *Gal3^−/−^* mice spontaneously show development of knee OA lesions with age.

Then, we compared the effect of mechanical overload, a major cause of OA, on WT versus *Gal3^−/−^* knee cartilages. Overload was induced by knee joint instability surgery (MNX) of 2 month-old mice. At 4 weeks after surgery, *Gal3^−/−^* cartilages displayed fissuration and proteoglycan loss (mean OARSI score 4.3 (range 1.0–9.0), *n* = 8 mice), whereas WT cartilages were essentially unaffected (mean OARSI score 1.6 (range 0–4.0), *n* = 8 mice, *p* = 0.014) (Figure 2A).

These results establish that mice lacking *Gal3* show severe OA during aging and in response to mechanical overload as compared with WT animals.

### 2.2. Deletion of GAL3 Increases Chondrocyte Catabolic Activity and Hypertrophy

Profound changes in chondrocyte metabolism are a hallmark of OA. To investigate the effect of GAL3 on the anabolic and catabolic activity of chondrocytes, we studied the expression of chondrocyte markers and different proteinases in two experimental settings: directly on cartilage sections and in primary cultures of articular chondrocytes from newborn mice.

First, we stained cartilage sections with antibodies directed against ADAMTS-5, one of the most potent aggrecanases involved in cartilage destruction [23]. The staining was greater in *Gal3^−/−^* than WT cartilages both during aging-related OA and after mechanical-induced OA. In 14-month-old mice (Figure 1B), this increase in ADAMTS-5 staining was seen in knee cartilages, as confirmed by comparing the percentages of ADAMTS-5+ chondrocytes (Table 1). In mechanical-induced OA (Figure 2B), the relative increase in ADAMTS-5+ chondrocytes was more pronounced in *Gal3^−/−^* than WT cartilages (Table 1). These data were completed by measuring the quantity of *Adamts-5* mRNA in cultured chondrocytes. We found a 60% increase of *Adamts-5* mRNA expression in *Gal3^−/−^* than WT cells without treatment (*p* < 0.05, *n* = 5 independent experiments), so the difference in level of ADAMTS-5 protein may result, at least in part, from differences at the transcriptional level. In line with the in vivo data showing an overproduction of ADAMST-5 in *Gal3^−/−^* OA cartilages, with IL-1β treatment, *Adamts-5* mRNA content was increased more in *Gal3^−/−^* than WT chondrocytes (4.2-fold vs 1.8-fold, *p* < 0.01, *n* = 5) (Figure 3A)

Second, we tested global metalloprotease (MMP) activity by VDIPEN staining of cartilage sections. The number of VIDPEN-positive cells was greater in *Gal3^−/−^* than WT cartilage (Table 1). In addition, the intensity of the VIDPEN staining, located in the pericellular matrix, was greater in *Gal3^−/−^* than WT cartilage (Figure 2B). In parallel to these in vivo data, IL-1β treatment induced significantly higher mRNA levels of *Mmp3* in *Gal3^−/−^* than WT chondrocytes (Figure 3B).

We next examined the expression pattern of type X collagen, a marker of the terminal (i.e., hypertrophic) stage of chondrocyte differentiation. After staining with anti-type X collagen serum, we observed a relatively weak signal, restricted to the calcified zone, in WT cartilages from old animals or mechanically overloaded animals. In contrast, the intensity of the staining was much stronger and expanded both cartilage zones in corresponding *Gal3^−/−^* cartilages (Figure 1B and Figure 2B). Consistently, the mRNA content of type X collagen as well as *Mmp13* was higher in *Gal3^−/−^* than WT chondrocytes (Figure 3C).

Finally, we assessed whether *Gal3* deletion altered chondrocyte anabolic activity. We observed the same expression pattern for type-2 collagen and aggrecan between WT and *Gal3^−/−^* cartilages after MNX. The staining intensity and proportion of positive-stained chondrocytes were identical between WT and *Gal3^−/−^* cartilages (Figure 4A). In vitro studies showed the same expression of *Col2a, Acan* and *Sox-9* between WT and *Gal3^−/−^* articular chondrocytes both at the basal level and after IL-1β stimulation (Figure 4B).

Hence, in the absence of *Gal3*, although chondrocyte anabolic activity was not altered, the overall catalytic activity of chondrocytes was greatly enhanced and the hypertrophic stage was much more prominent, which indicates an acceleration of the differentiation process.

To ensure that these effects were not secondary to a compensatory overexpression of other galectins, we compared the gene expression of different galectins between WT and *Gal3^−/−^* articular chondrocytes. The expression of *Lgal1*, *2*, *7* and *12* was similar between WT and *Gal3^−/−^* chondrocytes, whereas that of *Lgal4, 8* and *9* was slightly changed (1.27-, 0.78- and 0.86-fold, respectively) (Figure 4C).

### 2.3. GAL3 Deletion Increases Chondrocyte Apoptosis via the Mitochondrial Pathway

During OA, chondrocyte hypertrophy leads to chondrocyte death [24]. TUNEL assay showed a greater number of apoptotic cells in *Gal3^−/−^* than WT cartilages, both with aging and after MNX (Figure 5A, Table 1).

To determine which apoptotic pathway(s) was/were affected in the absence of GAL3, we treated primary cultures of articular chondrocytes with TNF-α (20 ng/mL), which triggered the extrinsic pathway, or with actinomycin D (ActD; 50 nmol/L), which triggered the intrinsic pathway. With ActD treatment, both the number of TUNEL-positive cells and level of Cas3 activity increased to a higher extent in *Gal3^−/−^* than in WT chondrocytes (*p* < 0.01) (Figure 5B,C). In contrast, the increase in Cas3 activity with TNF treatment was similar in *Gal3^−/−^* and WT chondrocytes. Of note, TNF had no detectable effect on the number of TUNEL-positive cells under these experimental conditions.

Taken together, these results suggest that in cultured chondrocytes, the extrinsic pathway of apoptosis is GAL3 independent while the intrinsic pathway is overstimulated in the absence of GAL3. This latter conclusion could be confirmed by directly monitoring the state of mitochondrial homeostasis. Indeed, we found that the cytochrome C release induced by ActD treatment was significantly more extensive in *Gal3^−/−^* than WT chondrocytes (Figure 5D).

### 2.4. GAL3 Deletion Induced Alteration of Chondrocyte Primary Cilia

As GAL3 is associated with the basal body of primary cilia in renal epithelial cells [22] and primary cilia was involved in apoptosis, we studied the primary cilia of WT and *Gal3^−/−^* mouse chondrocytes. With confocal microscopy, we could visualize the primary cilia of cultured WT chondrocytes stained with antibodies directed against the acetylated form of α-tubulin (Figure 6A). Double labelling experiments revealed that *Gal3* was highly enriched at the base of the cilium, where it colocalized with γ-tubulin, a marker of the basal body. We concluded that *Gal3* is present at the level of the basal body of primary cilia of chondrocytes (Figure 6A).

The total number of ciliated cells was lower in *Gal3^−/−^* than WT chondrocytes (59% vs. 68%); this difference was even greater in serum-free cultures (66% vs. 81%). We observed normal looking (i.e., “straight shaped”) primary cilia at the surface of 60% of WT chondrocytes (*n* = 642) but only 10% of *Gal3^−/−^* chondrocytes (*n* = 769); moreover, cilia were significantly longer in *Gal3^−/−^* than WT cultures (Table 2). We also found cells with stunted or curved primary cilia and occasional double-ciliated cells. Although these “abnormalities” were present in cultures of WT chondrocytes, they were far more abundant in mutant chondrocytes. Upon serum starvation, a condition that favors ciliogenesis, the ratios of aberrant primary cilia differed between WT and mutant chondrocytes (Table 2). The most frequent type of defect was the stunted shape, which was observed in 44.4% of *Gal3^−/−^* chondrocytes and only 16.4% of WT chondrocytes (*p* < 0.001) (Figure 6B).

Taken together, these data show that the primary cilia located at the surface of *Gal3^−/−^* chondrocytes are straight but are abnormally long or are misshapen.

## 3. Discussion

In this study, we reported that *Gal3^−/−^* mutant mice showed OA development in two different models and that GAL3 had major role in chondrocyte primary cilium formation. Classical changes in chondrocyte metabolism detected included enhanced catabolic activity, accelerated rate of differentiation and high apoptosis, with no change in anabolic activity. Furthermore, using the MNX OA model, at one month after surgery, *Gal3^−/−^* knees already displayed histological OA damages similar to that observed in older animals, whereas WT knees still appeared unchanged. Although a sham operation was performed in the contralateral legs, the contralateral knee cartilages of WT and *Gal3^−/−^* mice were normal 4 weeks after sham surgery. Taken together, our results further establish GAL3 as a major factor involved in maintaining cartilage homeostasis. Compared to our previous study where we only observed pre-OA signs in 4-month-old *Gal3^−/−^* mice, we clearly showed in this work cartilage destruction induced by aging and joint destabilization OA models [19]. While, in the former study, 4-month-old *Gal3^−/−^* mice only displayed cartilage structure parameter modifications without cartilage damage, we clearly showed, in this study, that 14-month-old *Gal3^−/−^* mice had severe OA cartilage destructions [19]. Moreover, while cartilage damages induced by mono-iodoacetate (MIA) injection were similar between *Gal3^−/−^* and WT mice, they were more severe in *Gal3^−/−^* mice than in WT mice when induced by joint instability [19]. Taken together, these results implicated GAL3 in the cartilage response to mechanical stress. This situation might depend in part on primary cilium functions.

Indeed, the primary cilium modulates several signaling pathways involved in chondrocyte maturation, survival, or death [25,26,27,28,29,30]. We previously observed that in renal and tracheal epithelia, a major site of GAL3 accumulation is the basal body of primary and motile cilia, respectively [22,31]. We report here that GAL3 is indeed highly enriched at the basal body of chondrocyte primary cilia and in mutant chondrocytes the primary cilia were misshapen or straight but significantly longer in mutant than WT cells. These results agree with the ciliogenesis defects observed in tracheal epithelial cells [21]. Because the primary cilium displayed mechanosensing properties in several cell types [6,22,31,32], its defects, consecutive to the absence of GAL3, might participate in OA development via an abnormal mechanical response. Structural abnormalities of primary cilium morphology altered cilium functions as described in *Bbs2^−/−^* and *Bbs4^−/−^* cilia which displayed a reduced cilium beat frequency compared to normal cilia [33]. Moreover, cilium length regulated anterograde and retrograde ITF transport, protein cargo traffic, protein signaling and mechanical transduction [34,35]. Primary cilium length was regulated by intracellular calcium, AMPc and protein kinase A activation, mechanical compression and osmotic challenge [7,9,34,35]. Increased cilium length accelerated anterograde ITF transport leading to increase calcium flux delivery [34]. The cilium length response created a negative feedback loop leading to cilium shortening and reduction of mechanical transduction [34]. Similarly, low-intensity ultrasound mechanical stimulation promoted elongation and bending of chondrocyte primary cilium and these length and shape changes were reversible [35]. Thus, the increased primary cilium length induced by GAL3 deletion might dysregulate chondrocyte responses to mechanical stimulation. Alternatively, misshapen primary cilia might increase chondrocyte apoptosis as described in tubular cells [36]. However, the direct link between these two results (misshapen and elongated primary cilium and OA phenotype) needs further study. We hypothesized that GAL3 might regulate Ihh signaling and chondrocyte hypertrophy differentiation and chondrocyte apoptosis [25,27,28,29,30,37].

In our original study, we observed accelerated and disorganized chondrocyte differentiation in the growth plate of *Gal3^−/−^* embryonic bones along with increased expression of Ihh in mature and hypertrophic zones, which raises the question of how GAL3 might impinge on cartilage physiology [38]. The present results reveal GAL3 involved in articular chondrocyte differentiation and death. The role of galectins in apoptosis has been extensively documented. Depending on the cell type and/or subcellular localization, GAL3 could be proapoptotic or antiapoptotic [30,39,40]. Here, we observed that cytosolic GAL3 prevented chondrocytes from mitochondrial-dependent apoptosis, which agrees with a previous study showing that GAL3 inhibited cytochrome C release from mitochondria via synexin interaction [41]. In this study, we could not exclude that extracellular GAL3 contributed to the OA phenotype induced by GAL3 deletion.

## 4. Materials and Methods

### 4.1. Animals and Protocol for OA-Induced Joint Surgery

Non-littermate wild-type (WT) and *Gal3^−/−^* 129 SvEV mice were maintained in a specific pathogen-free animal facility and handled according to the French regulation for animal care. *Gal3^−/−^* 129 SvEV mice have been generated by F. Poirier since 1998 [31]. Experiments followed the local Guidelines for Animal Experimentation (Ethics Committee Lariboisière-Villemin no.CEEALV/2012-02-01, Paris, France; approval date: 01 February 2012). OA was observed during aging (at 3 and 14 months) and on induction by partial resection of the medial meniscus (meniscectomy (MNX)) in the right knee of 2-month-old male WT (*n* = 8) and *Gal3^−/−^* (*n* = 8) mice as described [42,43].A sham operation was performed on the left knee. Surgery was carried out under sedation with ketamine/xylazine (160 and 6 mg/kg, respectively) and buprenorphin (50 µg/kg). Four weeks after MNX, mice were sacrificed by cervical dislocation and knee articulations were collected. OA lesions were analyzed according to OARSI recommendations [44]. Tissue samples were fixed in 4% paraformaldehyde (PFA) for 24 h, then soaked in 0.5 M EDTA at 4 °C for 10 days for complete decalcification before paraffin embedding. For the aging model, analysis involved 8 each of 3-month-old WT and *Gal3^−/−^* mice and 7 and 5 of 14-month-old WT and *Gal3^−/−^* mice, respectively.

### 4.2. Safranin O Staining and OARSI Scoring

Decalcified knee samples were embedded in paraffin. Sagittal sections 5-µm-thick were stained with safranin-O. Because OA lesions only occurred in the medial compartment in the MNX model, OA scoring was performed in sagittal sections of medial femorotibial joints. Double-blind (to surgery type and mouse genotype) OA grading was performed at three levels (2–3 slides/level) separated by a 50- to 60-µm interval by two researchers (N. Hafsia, HK Ea). Cartilage lesions in tibias and femurs were scored for OA on a scale from 0 to 12. The average of the worst total score observed by each researcher was used as the OA score for each mouse [44,45,46]. The kappa score of inter-observer agreement was 0.62 which corresponded to a strong correlation.

### 4.3. Antibodies

Primary antibodies were rat monoclonal antibody directed against GAL3 kindly provided by Dr. HakonLeffler (Lund University, Sweden), anti-VDIPEN IgG by Irwin Singer and Ellen Bayne (Merck Research Laboratories, Rahway, NJ, USA), mouse monoclonal anti-acetylated α-tubulin and rabbit polyclonal anti-γ-tubulin from Sigma-Aldrich (Saint-Louis, MO, USA), mouse monoclonal anti-cytochrome C (Bd Pharmingen, Le pont de Claix, France), mouse monoclonal anti-type X collagen (Diagomics, Blagnac, France), rabbit polyclonal anti-ADAMTS-5 (Abcam, Cambridge, UK), mouse monoclonal anti-type 2 collagen (Abcam), and rabbit polyclonal anti-aggrecan (Millipore, Guyancourt, France). Secondary antibodies were goat anti-mouse Alexa-488 and goat anti-rabbit Alexa-568 (Life Technologies, Paisley, UK).

### 4.4. Immunostaining of Knee Sections

Antigen retrieval in serial 5-µm-thick sagittal paraffin sections of decalcified knee samples was obtained by incubation with citrate buffer 10 mM, pH 6, for 4 h at 70 °C, then hyaluronidase (1 mg/mL, 37 °C, 15 min) (collagen 2, aggrecan, ADAMTS-5) or pepsin (0.1% in phosphate buffered saline [PBS], 37 °C, 2 h) (collagen X) was added. Sections were incubated with rabbit primary polyclonal antibodies against type 2 collagen (1/200, 2 µg/mL), aggrecan (1/200, 2.5 µg/mL) and mouse primary monoclonal anti-type X collagen antibody (1/100, 2 µg/mL) at 4 °C overnight. ADAMTS-5 and VDIPEN immunostaining was performed as described [42,47]. Nonspecific binding sites were blocked with MOM Blocking Reagent (Vector Labs, Peterborough, UK). Negative controls were non-specific IgG antibodies. For these antibodies, the number of positive staining cells was manually counted (4 fields per section, histolab software (Excilone, Elancourt, France) by two researchers (N. Hafsia, HK Ea) who were blinded to each other’s results. In case of discrepancy, a third analysis was performed by both researchers to reach consensus.

### 4.5. Primary Cultures of Chondrocytes

Articular chondrocytes were isolated from 6-day-old newborn mice as described [48]. WT and *Gal3^−/−^* chondrocytes were collected and pooled from a litter of WT and *Gal3^−/−^* pups, respectively. Chondrocytes were then plated (at least triplicate/condition) at 10^5^ cells/well in 24-well plates in DMEM 4 g/L glucose (Gibco, Les Ulis, France) containing 10% heat-inactivated fetal calf serum (FCS) (GE Healthcare, Illkirch, France), 4 mM glutamine (Gibco), penicillin (100 U/mL) (Gibco), and streptomycin (100 µg/mL) (Gibco). Chondrocyte cultures were kept until confluence. FCS starvation was performed 24 h before stimulation or fixation. For gene expression studies, confluent WT and *Gal3^−/−^* chondrocytes in serum-free medium were stimulated for 24 h with mouse recombinant IL-1β (10 ng/mL; 201-LB/CF; R&D Systems, Lille, France).

### 4.6. Immunostaining of Primary Chondrocytes

Before immunostaining with anti-GAL3 (1/100), anti-γ-tubulin (1/200), or anti-acetylated α-tubulin (1/200) antibodies, confluent chondrocytes were fixed in −20 °C precooled methanol for 5 min, permeabilized for 30 min in PBS containing 0.025% saponin and 1% BSA, then incubated with primary antibodies overnight at 4 °C in PBS containing 0.025% saponin and 1% BSA. For cytochrome c immunostaining, chondrocytes were fixed for 5 min in acetone at −20 °C, then for 15 min in 4% PFA, permeabilized for 30 min in PBS containing 0.02% Triton X-100 and 3% BSA, before incubation with anti-cytochrome c antibody (1:200, 2.5 µg/mL) for 1 h at room temperature. Incubation with secondary antibodies was performed for 1.5 h at room temperature. Nuclei were detected by Hoechst 33342 staining (Life Science, Villeneuve-La-Garenne, France).

### 4.7. Primary Cilium Analysis

For primary cilium detection, confluent articular chondrocytes were fixed, incubated with primary antibodies (anti-γ-tubulin and anti-acetylated α-tubulin), then secondary antibodies (Alexa 568 anti-rabbit and Alexa 488 anti-mouse, respectively, (Life Technologies, Paisley, UK). Confocal images were acquired on a Leica TCS SP5 microscope with x63 lens magnification (Leica Microsystems, Wetzlar, Germany). Six random fields were analyzed for each sample. Confocal images were assessed by two researchers (M. Forien, D. Delacour). Primary cilium analyses involved use of 3D software, IMARIS 7.0 (Bitplane, Zurich, Switzerland). Six independent experiments were performed. In total, *n* = 769 *Gal3^−/−^* and *n* = 642 WT chondrocytes were analyzed to describe cilium abnormalities.

### 4.8. RNA Isolation and RT-qPCR

RNA extracts were prepared from chondrocytes in culture by using TRIzol reagent according to the manufacturer’s instructions (Life Technologies, Paisley, UK), followed by a purification step with RNeasy Mini Kit (Qiagen, Courtaboeuf, France). cDNA synthesis involved the High Capacity kit (Fisher Scientific, Illkirch, France). Relative mRNA levels were evaluated by quantitative PCR analysis with LightCycler (Roche Applied Science, Meylan, France) and ABsolute Blue qPCR SYBR Green Mix (Fisher Scientific). HPRT6 mRNA level was a normalization control. Primer sequences of different genes are listed in Table 3. Primer sequences were designed using the PrimerBlast website then checked for self-annealing sites, 3’ complementarity, and potential hairpin formation using the Oligo Calc: Oligonucleotide Properties Calculator. Finally, the specificity of the primers was checked by doing an in silico PCR on the UCSC website. The efficiency of the primers was checked by doing a PCR with different quantities of cDNA (12.5, 25, 50, 125 and 250 ng) and the corresponding melting curves. Results of qRT-PCR were the mean of at least 3 to 5 independent triplicate experiments.

### 4.9. TUNEL Assay

Paraffin sections underwent TUNEL assay according to the manufacturer’s instructions (Millipore, Guyancourt, France). Sections were deparaffinised, rehydrated, treated for 15 min with 1 mg/mL proteinase K (Sigma-Aldrich) in PBS, then 8 min in 3% H_2_O_2_, then incubated for 4 min in equilibration buffer before adding TdT enzyme for 1 h at 37 °C. The reaction was stopped with Stop/Wash Buffer, and diaminobenzidine-conjugated anti-digoxigenin antibody was used to detect apoptotic cells. Sections were counterstained with methyl green.

Primary chondrocytes seeded on glass coverslips after fixation with acetone/methanol (vol/vol) for 5 min at −20 °C underwent TUNEL staining. Then chondrocytes were incubated with equilibration buffer for 10 s before adding TdT enzyme for 1 h at 37 °C. The reaction was stopped with Stop/Wash Buffer, and rhodamine-conjugated anti-digoxigenin antibody was used to detect apoptotic cells. Chondrocytes were counterstained with DAPI (Sigma-Aldrich) for cell counting. Ten random immunofluorescent images/slide were obtained and Axiocam software (Zeiss Microscopy, Marly Le Roi, France) was used for counting.

### 4.10. Induction of Apoptosis

After 5 days in culture, subconfluent chondrocytes were washed, starved overnight in serum-free medium, then cultured for 48 h in complete medium containing 20 ng/mL tumor necrosis factor α (TNF-α) (R&D Systems, Lille, France) or 50 nM actinomycin D (ActD) (Sigma-Aldrich) for inducing the extrinsic or intrinsic apoptotic pathway, respectively. Three independent quadruplicate experiments were performed.

### 4.11. Caspase3 Activity

After apoptosis induction, chondrocytes were incubated on ice for 30 min in lysis buffer (10 mM Tris pH 7.4, 200 mM NaCl, 5 mM EDTA pH 7.4, 10% glycerol and 1% NP40). Lysates were centrifuged at 10,000× *g* for 5 min, and supernatants were collected and stored at −20 °C. Caspase3 (Cas3) activity was determined as described [49], then normalized for protein contain.

### 4.12. Statistical Analysis

Data are reported as mean ± SEM or median and ranges. Medians (and ranges) are presented in a box plot where the box edges are the first and third quartiles, the line within the box represents the median and the lines outside the box represent the spread of the values. The non-parametric two-tailed Mann–Whitney *U* test was used for comparing groups. Statistical analysis involved use of StatView (SAS Institute). *p* < 0.05 was considered statistically significant.

## 5. Conclusions

We provide genetic evidence that GAL3 is a key regulator of cartilage homeostasis in mice, particularly in response to mechanical stress induced by joint instability. GAL3 deletion favors OA development and severity and disturbs chondrocyte functions including formation of primary cilium, catabolic activities and cell death. Some reports have implicated GAL3 in human OA, so further characterization of this OA animal model could lead to novel therapeutic strategies to tackle this debilitating disease.

## Figures and Tables

**Figure 1 ijms-21-01486-f001:**
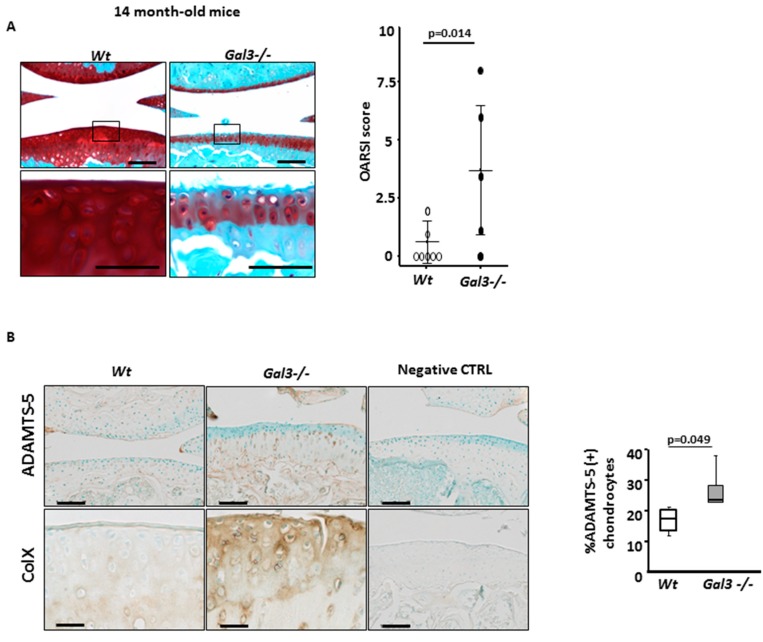
Galectin 3 (GAL3) deficiency induces osteoarthritis (OA) during aging. (**A**) Knee cartilage sections stained with safranin-O from non-operated *Gal3^−/−^* (*n* = 5 mice) and wild-type (WT) (*n* = 7 mice) 14-month-old mice. Low (top panels) and high (low panels) magnification are presented. OA lesions were assessed by OARSI score (right panel). (**B**) Immunohistochemistry of expression of ADAMTS-5 and type X collagen (*n* = 4 WT and *n* = 7 *Gal3^−/−^* mice). The percentage of ADAMTS-5-positive chondrocytes in *Gal3^−/−^* and WT cartilage was counted ((**B**), right panel). Negative controls were nonspecific IgG antibodies. Scale bars: 100 µm. Two-tailed Mann–Whitney *U* test between WT and *Gal3^−/−^.*

**Figure 2 ijms-21-01486-f002:**
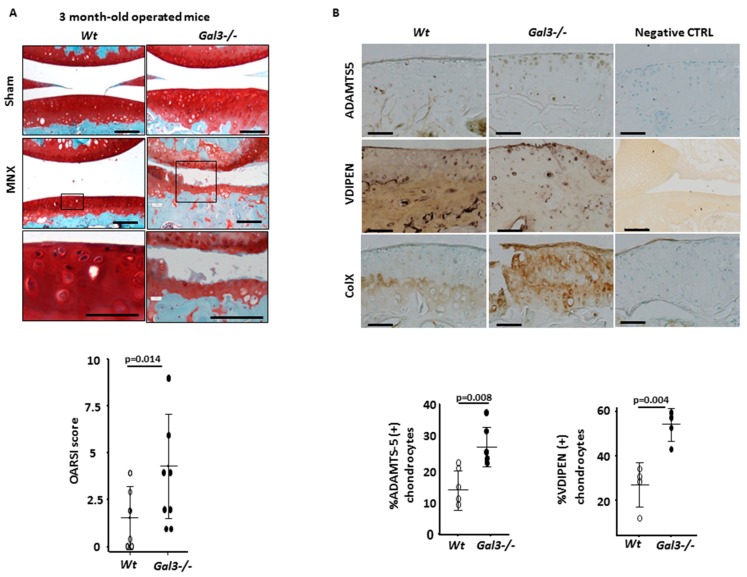
GAL3 deficiency induces OA during mechanical stress. (**A**) Knee cartilage sections stained with safranin-O after joint meniscectomy (MNX) or sham operation in 3-month-old mice (*n* = 8 mice for WT and *Gal3^−/−^*). Low (top panels) and high (low panels) magnification are presented. OA lesions were assessed by OARSI score (low panel). (**B**) Immunohistochemistry of expression of ADAMTS-5 (*n* = 5 WT and *Gal3^−/−^* mice), VDIPEN (*n* = 4 WT and *Gal3^−/−^* mice) and type X collagen (*n* = 8 WT and *Gal3^−/−^* mice). The percentage of ADAMTS-5– and VDIPEN-positive chondrocytes in *Gal3^−/−^* and WT cartilage was counted ((**B**), low panel). Negative controls were nonspecific IgG antibodies. Scale bars: 100 µm. Two-tailed Mann Whitney *U* test between WT and *Gal3^−/−^*.

**Figure 3 ijms-21-01486-f003:**
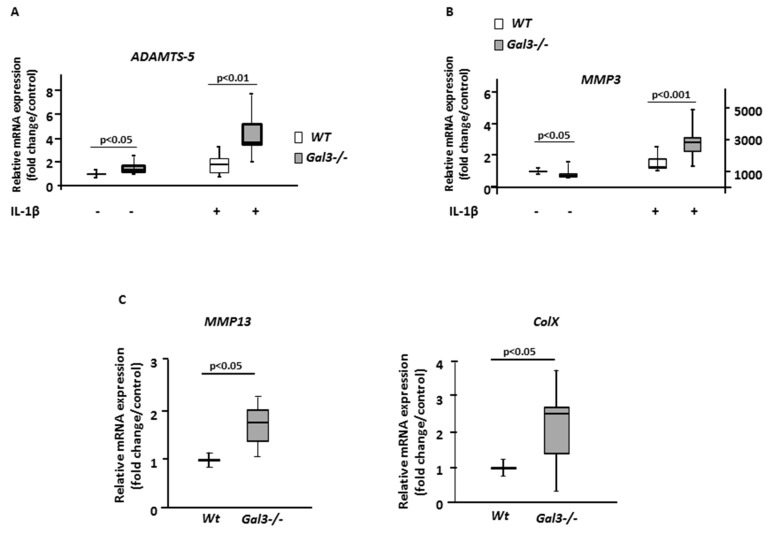
GAL3 deletion induces chondrocyte catabolism. qRT-PCR analysis of *Adamts-5* (**A**), *Mmp3* (**B**), *Mmp13* and type X collagen (**C**) mRNA expression in cultured chondrocytes from articular cartilage of *Gal3^−/−^* and WT newborn mice with and without IL-1β stimulation (10 ng/mL) (201-LB/CF, R&D Systems, Lille, France) (*n* = 5 independent experiments). Horizontal bars were medians, box edges are Q1–Q3 and whiskers are range. Two-tailed Mann–Whitney U test between WT and *Gal3^−/−^*.

**Figure 4 ijms-21-01486-f004:**
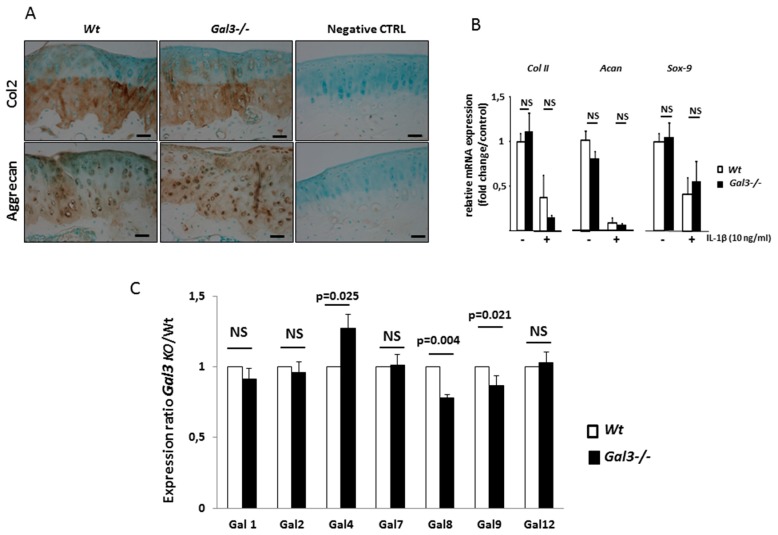
GAL3 deletion does not modulate chondrocyte anabolism. (**A**) Immunohistochemistry of expression of type II collagen and aggrecan in knee cartilage of WT and *Gal3^−/−^* cartilages after MNX (*n* = 5 of WT and *Gal3^−/−^* mice). Scale bars: 100 µm (**B**) qRT-PCR analysis of *Coll II*, *Acan and Sox9* mRNA expression in cultured chondrocytes from articular cartilages of *Gal3^−/−^* and WT newborn mice with and without IL-1β stimulation (10 ng/mL) (201-LB/CF, R&D Systems, Lille, France) (*n* = 5 independent experiments). (**C**) qRT-PCR analysis of *Lgals1*, *Lgals2*, *Lgals4*, *Lgals7*, *Lgals8*, *Lgals9* and *Lgals12* expression in cultured chondrocytes from articular cartilage of *Gal3^−/−^* newborn mice (*n* = 5 independent experiments). Two-tailed Mann–Whitney *U* test between WT and *Gal3^−/−^*.

**Figure 5 ijms-21-01486-f005:**
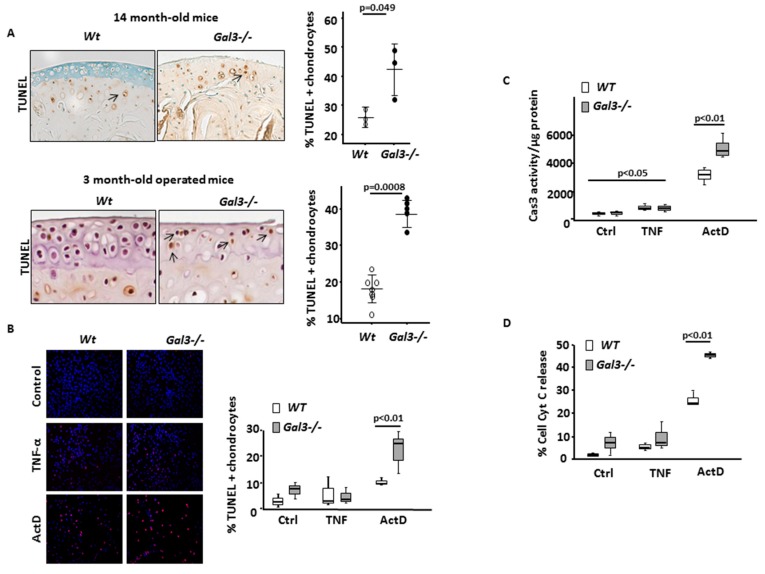
GAL3 deletion induces chondrocyte apoptosis via mitochondrial pathway. (**A**) Ex vivo, chondrocyte apoptosis was assessed by TUNEL labeling in 14-month-old mouse cartilages (*n* = 3 WT and *Gal3^−/−^* mice) (left and top panel) and 3-month-old mouse cartilages after MNX (*n* = 8 WT and *Gal3^−/−^* mice). (**B**) In vitro, articular cartilage chondrocytes from *Gal3^−/−^* and WT newborn mice were stimulated with TNF-α (20 ng/mL) or actinomycin (ActD, 50 nM) or left untreated (Ctrl) (*n* = 3 independent experiments). Percentage of TUNEL positive chondrocytes (A and B, right panels). (**C**) Caspase 3 (Cas3) activity (expressed in arbitrary units (AUs)) assessed 48 h after stimulation with TNF-α or ActD and normalized for protein contain. (**D**) Percentage of cytochrome c (Cyt c) release in chondrocyte cultures assessed by immunostaining and fluorescence microscopy. Scale bars: 100 µm. Horizontal bars were medians, box edges are Q1–Q3 and whiskers are range. Two-tailed Mann–Whitney *U* test between WT and *Gal3^−/−^*.

**Figure 6 ijms-21-01486-f006:**
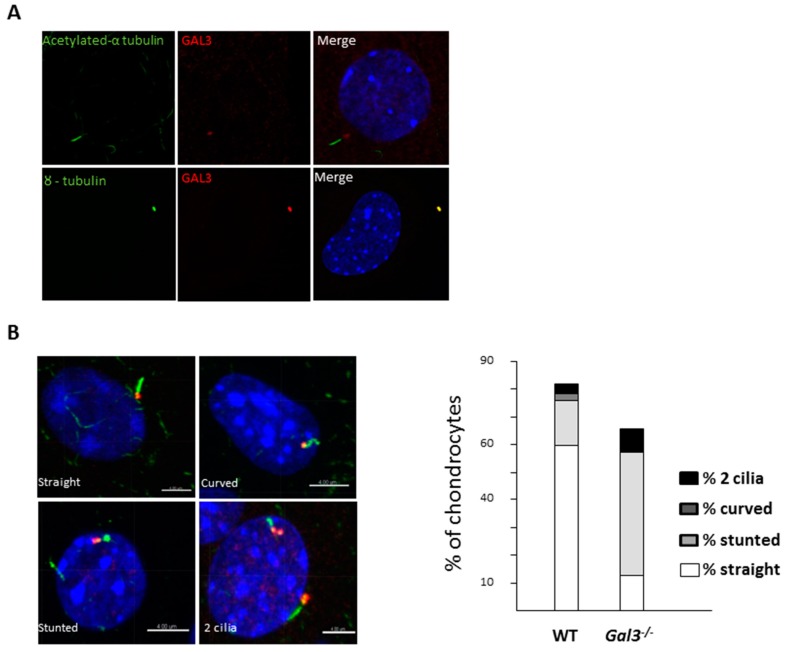
Deletion of galectin 3 (*Gal3*) induces alteration of articular chondrocyte primary cilium. (**A**) Confocal microscopy of acetylated α-tubulin (green) and GAL3 (red) (top pictures) or γ-tubulin (green) and GAL3 (red) (bottom pictures) in primary cilium of cultured wild-type (WT) chondrocytes (*n* = 5 independent experiments). (**B**) Various shapes of primary cilia were observed: straight, stunted, curved and double cilia. The relative proportions of each shape in WT versus *Gal3^−/−^* cultures are indicated. *n* = 642 WT and *n* = 769 *Gal3^−/−^* chondrocytes. Scale bars: 4 µm.

**Table 1 ijms-21-01486-t001:** Chondrocyte catabolism and apoptosis during age-induced and meniscectomy (MNX)-induced OA.

Catabolic and Apoptotic Changes	14-Month-Old Mice, Age-Induced OA			3-Month-Old Mice, MNX-Induced OA		
	WT	*Gal3^−/−^*	*p*	WT	*Gal3^−/−^*	*p*
%ADAMTS5 + cells	15.7 (9.3–17.7)	27.3 (23.5–55.9)	0.049	16.9 (11.8–21.0)	26.9 (21.0–38.1)	0.008
%VDIPEN + cells	NA	NA	NA	30.4 (12.6–34.7)	56.1 (44.0–58.9)	0.004
%TUNEL + cells	25.5 (22.5–29.5)	45.2(32.5–45.6)	0.049	18.3 (11.7–24.0)	39.6 (34.0–42.5)	0.008

The mean ratio [min, max] of femoral and tibia chondrocytes positively stained were shown. ADAMTS: a disintegrin and metalloprotease with thrombospondin-like repeat; VDIPEN: metalloproteinases-generated neoepitope; TUNEL: terminal deoxynucleotidyl transferase dUTP nick end labeling. NA: not assessed. Two-tailed Mann–Whitney U test between WT and *Gal3^−/−^*.

**Table 2 ijms-21-01486-t002:** Primary cilium anomalies in *Gal3^−/−^* chondrocytes.

Primary Cilium Anomalies	WT Chondrocytes	*Gal3^−/−^* Chondrocytes	*p*
Primary cilium length (µm)	1.76 (1.71–1.81)	1.93 (1.87–1.99)	0.0003
% abnormal primary cilia	12.4 (9.2–15.5)	29.4 (22.6–36.2)	0.03
% abnormal cilia (serum starvation)	21.7 (15.3–28.2)	53.9 (48.3–59.5)	0.001

Data were expressed as mean and 95% confidence intervals. Two-tailed Mann–Whitney *U* test between WT and *Gal3^−/−^.*

**Table 3 ijms-21-01486-t003:** Primer sequences of different genes.

Genes	Primer_F	Primer_R
*Hprt6*	GGTGGATATGCCCTTGACTATAATGA	CAACATCAACAGGAGTCCTCGTATT
*Acan*	CAG GGTTCCCAGTGTTCAGT	CTGCTCCCAGTCTCAACTCC
*Sox9*	GAAGCTGGCAGACCAGTACC3	GGTCTCTTCTCGCTCTCGTTC
*Col2a*	CCG TCATCGAGTACCGATCA	CAGGTCAGGTCAGCCATTCA
*ColX*	AAGGAGTGCCTGGACACAAT	GTCGTAATGCTGCTGCCTAT
*Mmp3*	ATGAAAATGAAGGGTCTTCCGG	GCAGAAGCTCCATACCAGCA
*Mmp13*	TGATGGCACTGCTGACATCAT	TGTAGCCTTTGGAACTGCTT
*Adamts-5*	TCAGCCACC ATC ACAGAA	CCAGGGCACACCGAGTA
*Lgals1*	CTCAAAGTTCGGGAGAGGT	CATTGAAGCGAGGATTGAAGT
*Lgals2*	CAGGGTCAGAGGTCAAGATCAC	GCCCACCCATGCTCAAGTAG
*Lgals4*	CATGCCTGAGCACTACAAGG	CGAGGAAGTTGATGGACTGAA
*Lgals7*	CCATGTCTGCTACCCAGCAC	CCTCACCGCATAGCAGGTTT
*Lgals8*	GGGTGGTGGGTGGAACTG	GCCTTTGAGCCCCCAATA
*Lgals9*	ATTCCAAATGGGCTTTACCC	AGGTGGAAAGCAATGTCACC
*Lgals12*	TCTGCATGCAAGGAGGTTTCA	GGCAACATCTGGCTGAGGAT

Acan: Aggrecan: Adamts-5; A Disintegrin And Metalloprotease with ThromboSpondin-like repeat; Col2a and ColX: type 2a and type X collagen; Hprt: Hypoxanthine-guanine phosphoribosyltransferase; Lgals: Galectins; Mmp: metalloproteases; Sox-9: SRY-box 9.

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
