# Peer review of "Galectin 3 Deficiency Alters Chondrocyte Primary Cilium Formation and Exacerbates Cartilage Destruction via Mitochondrial Apoptosis"

_ijms, 2020, doi:10.3390/ijms21041486_

Round 1
Reviewer 1 Report
While the team submits revised version together with a rebuttal letter. Unfortunately, the team seems to skirt around reviewer's comments and concerns over the osteoarthritis signs, which is unlikely prominent in the analysis the team has revealed.
It is well established that histopathology of OA signs become evident at 8 weeks postoperatively in experimental rodents. The team needs to make clear the experimental OA model they've done is relevant to cartilage biology or OA pathology, while the the KO mice show sever articular cartilage underdevelopment.
Again, the team seems to perform histomorphometry in cherry-picking way, For example, the tidemark and anatomical site are not consistent.
The image quality is not improved at all. The resolution and contrast of images are still too poor to be read.
Reviewer 2 Report
The change in the order of presentation of the data makes the manuscript more coherent and robust.
Author Response
We thank for your comments and time. We performed modifications as suggested. Please find below our responses to your concerns.
Reviewer 1.
I believe that this paper, showing high quality and original results, would benefit from another order of presentation of the results: first, in vivo data, then in vitro data; the in vitro data with the anabolism/catabolism studies showing that in vitro culture reflects very well in vivo results. This would strengthen the successive explorative study on the primary cilium in culture. The manuscript would end with this idea, exploiting in fact studies published in the literature, indicating links between primary cilium and apoptosis.
Answer: we are grateful for these comments.
Modification: we modified the manuscript as suggested by changing the order of presentation. The paper ended with primary cilium data.

Reviewer 3 Report
The manuscript title “Galectin 3 deficiency alters chondrocyte primary cilium formation and exacerbates cartilage destruction via mitochondrial apoptosis” Here Authors investigated the important of Galectin 3 (GAL3) in the formation of primary cilia, organelles that are able to sense mechanical stress and they evaluate the role of GAL3 in chondrocyte primary cilium formation and in OA in mice. Here Author induced OA by aging and partial meniscectomy of wild-type (WT) and Gal3-null 129SvEV mice (Gal3-/-). Furthermore, primary chondrocytes were isolated from joints of new-born mice and chondrocyte apoptosis was assessed by TUNEL labeling, caspase 3 activity, and cytochrome c release. Gene expression was assessed by qRT-PCR. In the results, the authors showed that GAL3 was localized at the basal body of the chondrocyte primary cilium. Primary cilia of Gal3-/- chondrocytes were frequently abnormal and misshapen. The deletion of Gal3 triggered premature OA during aging and exacerbated joint instability-induced OA. In both aging and surgery-induced OA cartilage, levels of chondrocyte catabolism and hypertrophy markers and apoptosis were more severe in Gal3-/- than WT samples. In vitro, Gal3 knockout favored chondrocyte apoptosis via the mitochondrial pathway.
The article is well written and the author did enough experiments to prove GAL3 is a key regulator of cartilage homeostasis and chondrocyte primary cilium formation in mice and deletion of Gal3 promotes OA. It was convincing enough. I would prefer this article is suitable for publication.